# Photoluminescent Polymer Aerogels with R, G and B Emission

**DOI:** 10.3390/ijms232416004

**Published:** 2022-12-15

**Authors:** Loredana Stan, Teodor Malutan, Irina Volf, Marcel Popa, Camelia E. Tincu, Corneliu S. Stan

**Affiliations:** 1Faculty of Chemical Engineering and Environmental Protection, Gh. Asachi Technical University, D. Mangeron 73 Ave., 700050 Iasi, Romania; 2Academy of Romanian Scientists, Ilfov Street, 077160 Bucharest, Romania

**Keywords:** polymer complexes, polymeric aerogels, photoluminescent materials, porous materials

## Abstract

In this work, three new polymer aerogels based on 2-hydroxy ethyl methacrylate (HEMA) complexes with Eu(III), Tb(III) and La(III) are prepared and investigated. The polymer aerogels present strong photoluminescence with emissions located in the red, green and blue regions of the visible spectrum. Depending on the water content used during the preparation path, the consistency of the photoluminescent aerogels varies from rigid, regularly shaped monoliths to a flexible, fibrous material with very low density. The morpho-structural investigation was performed by FT-IR, XPS and SEM. Thermal behavior was also evaluated, while steady-state fluorescence spectroscopy, absolute PLQY and lifetime were used for the investigation of their luminescent properties. The impressive photoluminescent emission located in the red, green and blue areas of the visible spectrum is preserved irrespective of the selected porosity. Their photo-emissive properties, tunable porosity and the convenience of the preparation path could be some arguments for applications as photonic conversion mediums in special-purpose optoelectronic devices or sensors.

## 1. Introduction

To date, a number of studies have been reported on the production of aerogel based on poly(2-hydroxyethyl methacrylate) (p-HEMA), most of them targeting medical applications: polymeric microporous aerogels prepared through cryo-polymerization based on HEMA and MBAA (*N*,*N*′-Methylene bisacrylamide) [1] meant for the selective recognition and purification of proteins; HEMA-based cryogels/aerogels with DNA introduced into the structure are used to remove anti-dsDNA antibodies from the blood plasma [2]; cryogels obtained by radical polymerization from HEMA, glycidyl ether and MBAA have been studied as support for controlled release drugs [3] or for the elimination of cholesterol from the coronary arteries [4]. Another medical application for the treatment of epidermal and transdermal wounds involves the use of polymeric composites prepared by embedding tannic acid into macroporous matrices of p-HEMA cross-linked with poly(ethylene glycol diacrylate) [5]. Biodegradable thermosensitive cryogels based on HEMA-lactate-dextran-*co-N*-isopropylacrylamide (NIPAM) have been reported for applications in bone tissue medicine [6], and p-HEMA-co-NIPAM, to obtain materials with augmented water retention in their porous structure [7]. Meant for use in medical devices, spongy monolithic cryogels have been prepared by radical polymerization of HEMA at −12 °C [8]. Also, environmental engineering applications were proposed for cryogel-based poly-HEMA: nicotinamide-modified HEMA aerogels have been studied to remove pesticides from wastewater [9]. In the same context, a series of HEMA-based aerogels have been tested for the retention in their porous structure of some heavy metal ions (cadmium and uranium) from the dispersion media [10,11]. Polymer composites based on poly-HEMA-GMA (glycidyl methacrylate) and silver nanoparticles have been studied and tested for water treatment for the removal of organic pollutants and microorganisms [12]. An interesting approach for advanced wastewater treatment consisted of the embedment of ZnO nanoparticles with photocatalytic action in the presence of graphene oxide in a p-HEMA matrix resulting in photochemically renewable composites [13]. Aerogels of various porosities based on p-HEMA have been evaluated in a recent study [14] for possible uses in air filtration systems. p-HEMA hydrogels were studied for obtaining disposable amperometry sensors for nitrite detection in water [15]. Because of their unique and remarkable luminescent properties, trivalent lanthanide cations remain extensively studied for applications in optoelectronic devices [16], telecommunications, sensors, biomedical imaging [17,18] and solar energy conversion [19]. Often, their luminescent complexes with various ligands have been incorporated into various host matrices, with particular attention being paid to polymer matrices [20,21]. Tb(III) cations were introduced into poly(vinyl alcohol) (PVA) matrices, resulting in monoliths with intense emission at 543 nm due to the ^5^D_4_→^7^F_5_ transition used in the preparation of doped monolithic silica glass [22]. An interesting reported approach was the preparation of RGB photoluminescent nanocomposites by introducing In(III), Eu(III) and Tb(III) complexes with terpyridine in silica matrices stabilized with poly(ethylene glycol) (PEG) prepared by a sol-gel process [23]. White photoluminescent emission was reported in polymer composites thin films by introducing Eu(III) and Tb(III) complexes into a poly(methyl methacrylate) (PMMA) matrix for application as photonic conversion media in PC-LED [24].

In this work, three new photoluminescent polymer aerogels with tunable porosity [25,26] were prepared and morpho-structurally investigated. The approach is based on 2-hydroxy ethyl methacrylate (HEMA) complexes with Eu(III), Tb(III) and La(III), which are cryo photo-polymerized and crosslinked with *N*,*N*′-Methylene bisacrylamide (MBAA). The prepared polymer aerogels present strong photoluminescence with emissions located in the red, green and blue regions of the visible spectrum. Due to their particular characteristics, these polymer complexes aerogels could be further developed in new approaches particularly relevant for various applications ranging from catalysis [27], water or air purification [28] to hydrogen storage [29] by using suitable cations coordinated with HEMA which are further processed in porous materials. Also, their MOF-like configuration [30,31] could also be an important argument for further developments and applications.

## 2. Results and Discussion

Figure 1 presents the complexation reaction between HEMA and *M*Cl_3_ (*M* = Eu, Tb, La) followed by photo-polymerization and MBAA cross-linking of the resulting complex [*M*(HEMA)_3_(H_2_O)_3_]. The coordination number of the central cation is nine through three covalent bonding with the oxygen in the -OH group and six coordinative bonds (three water molecules in the first coordination sphere and three coordinative bonding with the oxygen atoms within the carbonyl groups).

### 2.1. Thermal Stability Investigation

The mass loss diagrams for each prepared aerogel are presented in Appendix A. As could be noted, the behavior at thermal exposure of each of the three prepared aerogels is similar. The slight mass loss recorded in the 50–150 °C range could be attributed to the loss of remnant water content and, in the upper part of the interval, to the commencing of the degradation processes, followed in the 150–450 °C range by a steep mass loss due to the advanced degradation processes accompanied by volatile exhausting. From the potential application’s perspective, ~100 °C should be the upper limit in order to preserve the integrity of the prepared aerogels.

### 2.2. FT-IR Analysis

The recorded FT-IR spectra are presented in Figure 2. To highlight the spectral modifications occurring through complexation, the spectrum of the HEMA ligand is also included (Figure 2a). As could be noted in the case of all complexes, the established covalent bonding with the trivalent cation is present in the typical lower region of the spectra [32] with new occurring low-intensity peaks (475 cm^−1^ in the case of La(III), 480 cm^−1^ for Tb(III) and 481 cm^−1^ respectively, in case of Eu(III)). The coordinative bonding established between the central cation and the oxygen in the carbonyl groups is also clearly highlighted. The 1728 cm^−1^ peak specific to the stretching vibration of the carbonyl groups within the non-complexed HEMA ligand appears to split in two and is slightly displaced in the case of the complexes. Thus, the established coordinative bondings are responsible for the split shoulder configuration occurring at 1656, 1663 and 1666 cm^−1^, while the main peaks (1726, 1724 and 1727 cm^−1^ are slightly displaced due to the re-arrangements occurring through the vicinity of the central cation.

### 2.3. XPS Investigation

In Table 1 are detailed the overall concentrations of the Eu, Tb La trivalent cations, C and O, resulting from the wide XPS spectrum recorded for each of the prepared aerogels (Appendix A). The high-resolution O1s and C1s spectra recorded for each type of aerogel (Appendix A) revealed the relative concentrations of several types of functional groups/specific bonding, and the results are detailed in Table 2.

The XPS investigations revealed the establishment of the covalent bonding between the central cation (Eu(III), Tb(III) and La(III)) and the oxygen atom and also the coordinative bonding between the same central cations with the oxygen atoms within the carbonyl groups. The recorded results are in good agreement with the FT-IR investigation results.

### 2.4. SEM Investigation

The recorded SEM micrographs of the prepared aerogels are presented in Figure 3. In all three cases, it was revealed a large continuous interconnected pore structure with a sponge-like configuration. As can be noted, the morphology of all three aerogels is similar as a result of their structural resemblance. For a better view of the porous structure in Appendix A, SEM micrographs recorded at higher magnification (1000×) are presented. The aerogels present a moderate elasticity which can be easily compressed manually to remove, for example, the accumulated water inside the pores. Then, if the compressed aerogel is submerged in water for a few seconds, it easily returns to its original shape and size. Their behavior is similar to other related HEMA cryogels/aerogels [25].

### 2.5. Fluorescence Investigation

The prepared aerogels present intense photoluminescence emission with Eu(III) and Tb(III) typical narrow emission peaks. The favorable structure of the p(HEMA) ligand provides favorable sensitization of the metal-centered radiative transitions. The mechanisms that govern the radiation processes in the prepared aerogels can be grouped into two categories: (1) in the case of Eu(III) and Tb(III) aerogels, photoluminescence emission is due to the radiative transitions occurring in the lanthanide cation which constitutes the luminescent center as a result of a “classic” sensitization of the radiative transitions due to the “antenna” effect provided by ligand over the Eu(III) and Tb(III) cations [33]; (2) in the case of La(III) aerogel the emission occurs due to the influence of the central La(III) cation (heavy atom influence) over the excited states of the ligand [34]. In Figure 4a–c are presented the recorded excitation/emission spectra of the prepared aerogels. In the case of the Eu(III) aerogel (a), the emission spectrum was recorded at 360 nm excitation, the most significant emission peaks are located at 577, 590 and 615 nm due to the ^5^D_0_→^7^F_0_, parity allowed magnetic dipole ^5^D_0_→^7^F_1_, and respectively the electrical dipole allowed hypersensitive ^5^D_0_→^7^F_2_ transitions. The highest intensity peak is located at 615 nm due to ^5^D_0_→^7^F_2_ transition, which is strongly affected by the surrounding symmetry degree [35]. In most cases, the intensity of this peak increases in highly disordered surroundings, the ratio between the intensities of the 615 nm and 590 nm peaks being significant for the evaluation of the disorder/asymmetry of the surroundings of the central Eu(III) cation [36]. In the case of the Tb(III) aerogel (360 nm excitation), the most intense emission peak is centered around 542 nm, which is specific for the ^5^D_4_→^7^F_5_ transition, while the lower intensity peak located at 488 nm being due to the ^5^D_4_→^7^F_6_ transition [37]. The highest intensity excitation peak is located at 395 nm (within the UV-A range) in the case of the Eu(III) aerogel, while in the case of Tb(III) aerogel, the wider band’s highest intensity peak is located at 315 nm (UV-B range).

In the case of the La(III) aerogel (Figure 4c; 360 nm excitation), the intense blue emission is most probably due to the radiative transitions within the ligand as a result of the vicinity influence of heavy La(III) cations over the excited states occurring within various groups. Interestingly, the same phenomenon was also highlighted in the case of Y(III) cations, where excitation-dependent blue emission was recorded [38], the most notable differences being the location of the emission peaks and a slightly lower PLQY in the case of the La(III) aerogel. In these cases, the emission peaks are markedly broader, for the La(III) aerogel being located at 434, 448 and 455 nm. The emission peak is excitation wavelength dependent which could be an argument in favor of heavy atom vicinity influence over various emissive functional groups within the ligand since both La(III) and Y(III) cations are not known to present specific radiative transitions as in case of Eu(III) or Tb(III). For the La(III) aerogel, the broad excitation peak is located at 380 nm (UV-A range). In Table 3 are presented the recorded absolute PLQY for each prepared photoluminescent aerogel.

In Figure 5a,b are presented the excited states’ lifetimes for each group of the prepared aerogels. In the case of Eu(III) and Tb(III) aerogels, where the radiative transitions occur in the lanthanide cation due to the sensitization induced by the surrounding ligands, the dominant value is τ2 = 590 μs (96.93%) which is typical for lanthanide-containing photoluminescent compounds [39], while the τ1 = 15.14 μs (3.07%) is most probably due to the excited states of the ligand.

In the case of the La(III) aerogel, the recorded lifetime values are markedly shorter due to specific emission occurring due to the influence of the heavy central cation influence over the excited states of the ligand. Both recorded values (τ1 = 1.016 μs and τ2 = 9.18 μs) are both a result of the radiative transitions occurring within various groups of the ligand as a result of the vicinity of the La(III) cation. As could be noted, the dominant 1.016 μs (65.66%) value is comparable with the ligand-specific lifetime (15.14 μs −3.07%) recorded for the Tb(III) aerogel while the less present 34.34% (9.18 μs) is also in the same (two digits μs) range. In the case of Eu(III)/Tb(III) aerogels, the dominant lifetime is more than 50x longer compared to La(III). As presented above, the results are a consequence of the different mechanisms which govern the radiative transitions occurring in Eu(III)/Tb(III) aerogels vs. La(III) aerogel [40].

## 3. Materials and Methods

### 3.1. Materials

2-Hydroxyethyl methacrylate (HEMA) with 99% purity was purchased from Sigma-Aldrich Europium(III) chloride hexahydrate, EuCl_3_x6H_2_O (99.9%), Terbium(III) chloride hexahydrate, TbCl_3_x6H_2_O (99.9%) and Lanthanum(III) chloride hydrate, LaCl_3_xH_2_O (99.9%) were purchased from Alfa Aesar. N, N’-Methylene bisacrylamide (MBAA), C_7_H_10_N_2_O_2_, as the crosslinking agent (99%), and 1-Hydroxycyclohexyl phenyl ketone, HOC_6_H_10_COC_6_H_5_ (HCPK) as photo-initiator was also provided by Merck Millipore RO. All aqueous solutions were freshly prepared using ultra-pure distilled water (Millipore-Direct Q).

### 3.2. Methods

The Infrared (FT-IR) spectra were recorded in the 400–4000 cm^−1^ range using a Shimadzu IR Affinity 1S spectrometer according to the KBr method. The thermal stability was studied on a Mettler Toledo TGA-SDTA851e under an airflow rate of 20 mL/min. The heating rate was adjusted to 10 °C/min in the 50–900 °C range. The X-Ray Photoelectron Spectroscopy (XPS) analysis was performed on a KRATOS Axis Nova, using AlKa radiation with a 20 mA current and 15 kV voltage. The incident X-ray beam was focused on a 0.7 mm × 0.3 mm area of the surface. Wide XPS spectra were collected in the range of −10 to 1200 eV with a resolution of 1 eV and a pass energy of 160 eV. The high-resolution spectra were collected using a pass energy of 20 eV and a step size of 0.1 eV. SEM micrographs were recorded with a Hitachi SU-1510. Excitation/emission spectra and the absolute photoluminescence quantum yields (PLQY) were recorded using a Horiba Fluoromax 4P fluorescence spectrophotometer equipped with a solid-phase probe adapter and the Quanta-φ integration sphere. The PLQY measurements were performed according to the equipment manufacturer’s procedure, the values being calculated by the provided fluorescence software (FluorEssence ver.3.5.1.20). Excited state lifetimes were investigated using an Edinburgh Instruments FLS980 photoluminescence spectrometer. The double exponential function, which was used to fit the fluorescence lifetimes of the samples, is given by the equation: $I\left(t\right)=I_{1}\cdot e^{\left(-t/\tau_{1}\right)}+I_{2}\cdot e^{\left(-t/\tau_{2}\right)}$ where $\tau_{1}$, $\tau_{2}$ are the two characteristic lifetimes and $I_{1}\;\mathrm{and}$
$I_{2}$ are their relative amplitudes.

### 3.3. Synthesis of p-(HEMA-M(III)) Aerogels

In the first step, complexes of HEMA and Eu(III), Tb(III) and La(III) were prepared at a 1/3 cation/ligand ratio according to the procedure described in a previous study [38]. Briefly, 1 mmol of cation salt (*M*Cl_3_) was reacted with 3 mmol of HEMA in an aqueous medium at 45–50 °C under moderate stirring for about 4–5 h. To avoid the presence of unreacted byproducts, in practice, the 1/3 cation/ligand ratio will be altered slightly in favor of the ligand. In the second step, the resulting aqueous complex aqueous solution was mixed with the cross-linker (MBAA) and polymerized in the presence of the photo-initiator (HCPK). The polymerization process takes place at −10/−12 °C for 24 h in the presence of UV-B (310 nm) radiation provided by an 8 W lamp. The cryo-polymerization process takes place in a conveniently chosen UV transparent recipient according to the intended final shape of the aerogel. Then, in the third step, the resulting cryogel is freeze-dried, thus, resulting in the aerogel. By varying the amount of water in the mixture prior to the photopolymerization, the resulting aerogel achieves a configuration ranging from a denser monolithic to a very low-density fibrous configuration (Figure 6a,b). In the same Figure 6 are presented some samples of the prepared photoluminescent aerogels under UV (370 nm) excitation provided by a regular UV laboratory lamp. It is worth mentioning that prior to photo-polymerization, the PL emission is not present in the initial aqueous solution consisting of the HEMA-*M*(III) complex and the cross-linker, most probably due to the vibrational deactivation path provided by the presence of the -OH groups within the surroundings of the central cation.

## 4. Conclusions

The paper reports three new aerogels with tunable porosity and excellent photoluminescent properties. Their preparation path involves the preliminary complexation between HEMA and the trivalent Eu(III), Tb(III) and La(III) cations. The aqueous solutions of the complexes are photo-polymerized at low temperatures in the presence of o crosslinker and subsequently freeze-dried to result in highly porous materials. Depending on the water content of the complex solutions prior to cryo-polymerization, the morphology of the resulting aerogels could be tuned from monolithic, higher density to fibrous, very low-density configuration. The prepared aerogels were morpho-structurally investigated in detail through FT-IR, XPS, SEM and fluorescence spectroscopy. From the practical point of view, their tunable porosity, ease of preparation, and remarkable photoluminescent emission could be notable arguments in favor of applications ranging from sensors to various optoelectronics devices. A suggested possible application is in the moisture content surveillance in air monitoring or air conditioning/transportation systems due to the specific change of PL emission intensity of the Eu(III) or Tb(III) complexes in the presence of water. Also, due to their facile configuration in various shapes, they could be interesting for applications as photonic conversion mediums in various optoelectronic devices. In conjunction with UV LED arrays which provide the required excitation, they could be easily configured as multicolor lighting sources for commercial or architectural illumination.

## Figures and Tables

**Figure 1 ijms-23-16004-f001:**
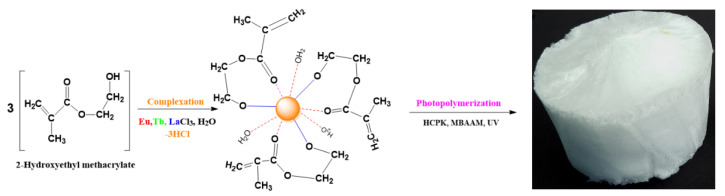
Complexation reaction and the obtained aerogel after photopolymerization.

**Figure 2 ijms-23-16004-f002:**
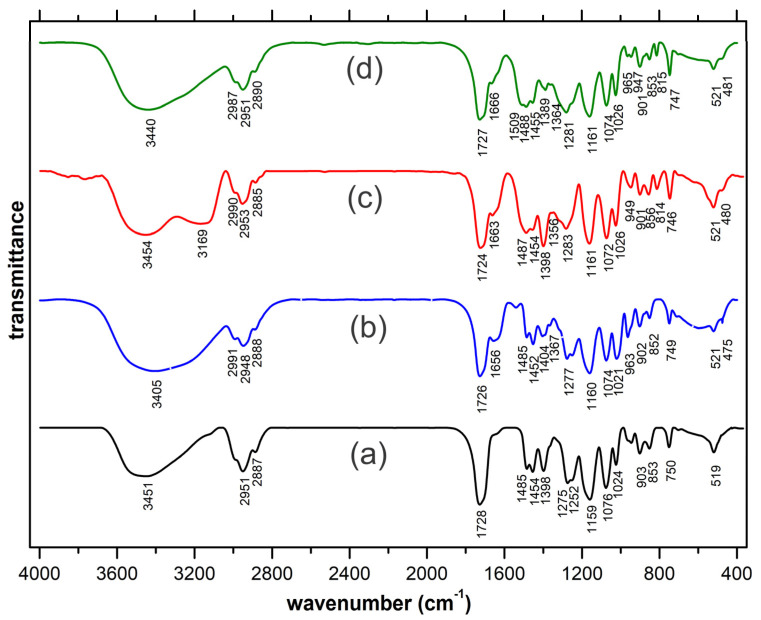
Spectra FT-IR spectra were recorded for: (**a**) HEMA ligand, (**b**) La(III), (**c**) Eu(III), and (**d**) Tb(III) aerogels.

**Figure 3 ijms-23-16004-f003:**
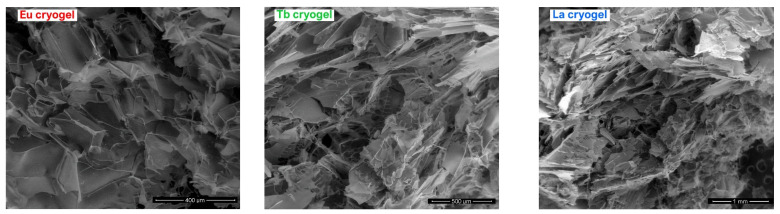
SEM micrographs of the prepared photoluminescent aerogels.

**Figure 4 ijms-23-16004-f004:**
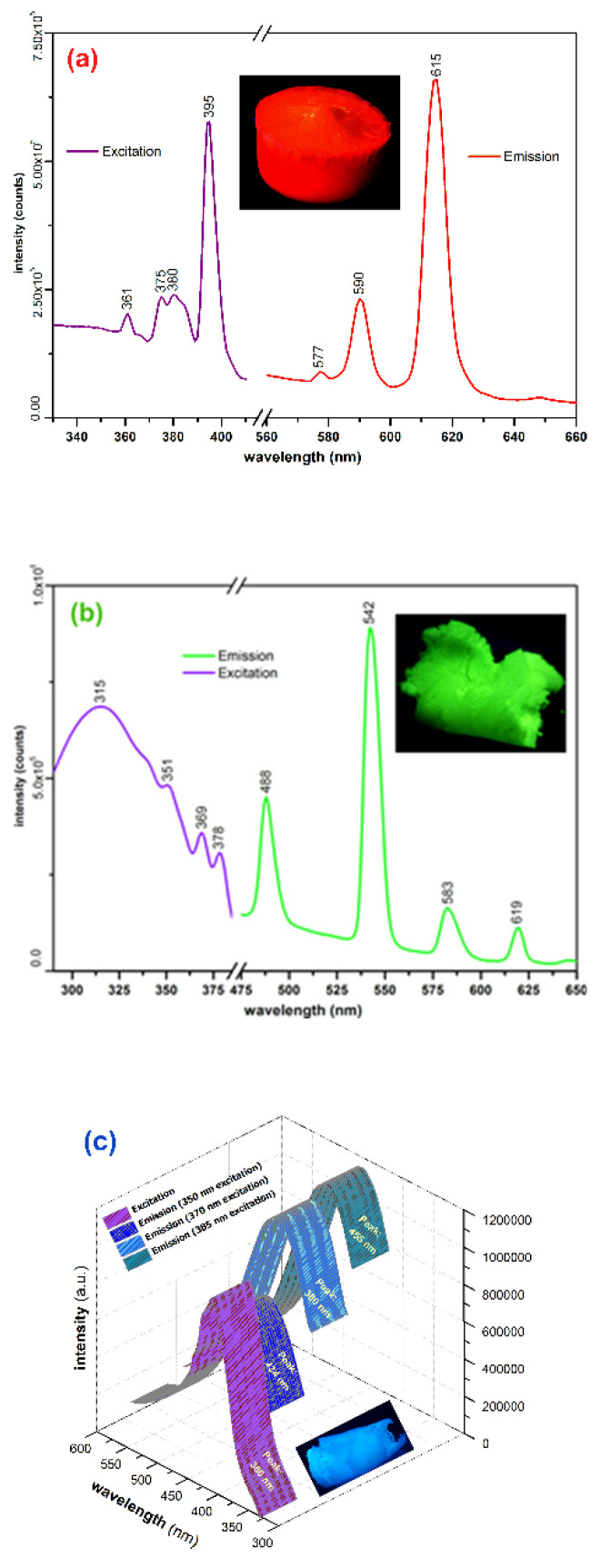
Excitation/emission spectra of prepared aerogels: (**a**) Eu(III) aerogel, (**b**) Tb(III) aerogel, (**c**) La(III) aerogel.

**Figure 5 ijms-23-16004-f005:**
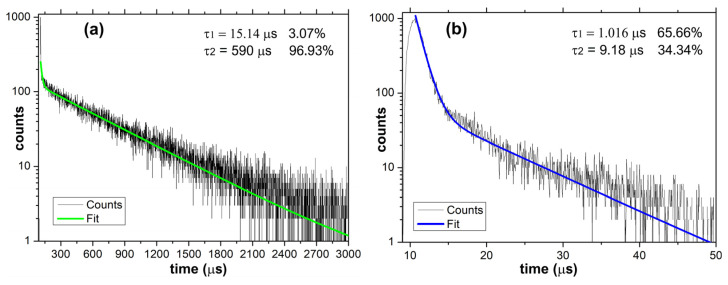
Luminescence lifetimes were recorded for the (**a**) Tb(III) aerogel and (**b**) La(III) aerogel.

**Figure 6 ijms-23-16004-f006:**
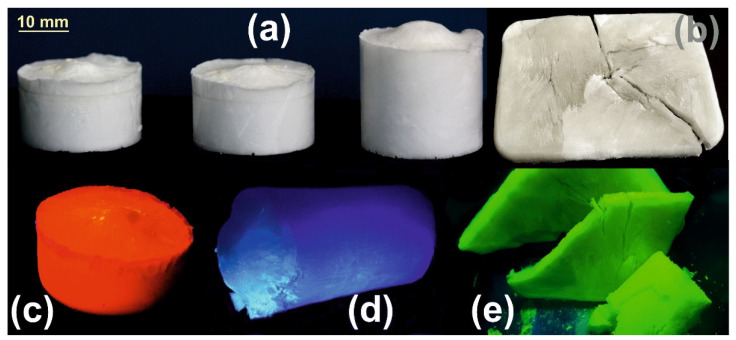
Prepared photoluminescent aerogels (**a**) Regular shape denser monolithic, (**b**) lower density fibrous configuration under ambient lighting, (**c**) Eu(III), (**d**) La(III) and (**e**) Tb(III) aerogels under 370 nm UV excitation.

**Table 1 ijms-23-16004-t001:** C, O, Eu(III), Tb(III), and La(III) concentrations recorded for the prepared aerogels.

Polymer Complex	Eu(III) Aerogel			Tb(III) Aerogel			La(III) Aerogel		
Element	*C*	*O*	*Eu(III)*	*C*	*O*	*Tb(III)*	*C*	*O*	*La(III)*
Atomic concentration (%)	80.8	18	1.2	80.5	18.2	1.3	79.5	19	1.5
Mass concentration (%)	59	18	23	67	19	14	67	21	12

**Table 2 ijms-23-16004-t002:** The concentrations of various groups/bondings within each prepared aerogel according to the high-resolution C1s and O1s spectra.

Polymer Complex	Eu(III) Aerogel			Tb(III) Aerogel			La(III) Aerogel		
*C1s high-resolution spectra*									
**Group/Bonding type**	O-C	O=C	C-C/C-H	O-C	O=C	C-C/C-H	O-C	O=C	C-C/C-H
**Mass concentration (%)**	12.4	6.9	80.7	11.8	7	81.2	19	10	71
*O1s high-resolution spectra*									
**Group/Bonding type**	O-C	O=C	O-Eu	O-C	O=C	O-Tb	O-C	O=C	O-La
**Mass concentration (%)**	30.4	65.7	3.9	28.5	67.5	4	38.6	49.3	12.1

**Table 3 ijms-23-16004-t003:** Absolute PLQY recorded for the investigated aerogels.

Aerogel	Absolute PLQY (%)/Excitation WAVELENGTH (Nm)	Relative Error (+/−)
Eu(III) aerogel	42/395	0.022
Tb(III) aerogel	36/315	0.020
La(III) aerogel	26/380	0.025

## Data Availability

The data presented in this study are available on request from the corresponding author.

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
