# Peer review of "Photoluminescent Polymer Aerogels with R, G and B Emission"

_ijms, 2022, doi:10.3390/ijms232416004_

Round 1

Reviewer 1 Report

This manuscript reports on an important topic, the results are interesting and the methods are adequate, so it should be published.

There are few comments for a minor revision:

1. Spelling should be checked, I have found, for example:

line 16: "Lifetime" with a capital "L" - should not it be "l"?
lines 234 and 245: Dots in the end of sentences are missed.

Also, I am not sure about many commas and article choices, please, consider a verification of the spelling.

2. Figure 4 presents the excitation spectra for all aerogels, however, the PL spectra are presented only for the excitation wavelength of 360 nm (350, 355 and 370 for the La aerogel). Why 360 +-10 nm were chosen? It seems not optimal for any of the aerogels (the optimal is 395 nm for Eu, 315 nm for Tb, and 380 nm). Moreover, the La aerogel PL spectra show that even in this short range (350-370 nm) the spectrum is very sensitive to the excitation wavelength - so, how we can consider the reported spectra representative for the aerogels luminescence properties?

If it is not very difficult for the authors, I would recommend to add spectra at different excitation wavelengths (ref. to the image in the file attached). PL data on the ligand reference (w/o a metal complex) is also desirable.

3. The tau_2 time in Fig. 5b seems to be not really correct (the fitting curve is significantly lower than the experimental data). 

4. Figure 6 would be more informative, if the it has a scalebar. And why the excitation wavelength of 370 nm was chosen, when the authors report 360 nm PL spectra in Figure 4?

5. Is there any information about PL properties of squeezed and refilled with water aerogel samples? It especially actual for the La(III) aerogel, since the Authors claim that the PL properties are governed by the surroundings of the La atom (lines 170-172). What is the stability of the synthesized aerogel samples, do they reproduce the spectra/PLQY after several weeks/months? Are they suitable for the outdoor applications?

6. SEM images in Fig. 3 have the resolution that does not allow to see the porous structure. The images like the SEM in the Supporting Information are more desirable. What was the electron beam energy (in keV)?

Author Response

Dear Reviewer,

Thank you for your constructive comments, we highly appreciated the spirit on which these comments were formulated. Please see the attachment (in pdf format) which include our point to point reply to the comments. However, we do apologize for our long reply at comment 2 but given the interesting point raised by the reviewer, we found very useful to extend the opinions exchange and debate over this particular matter.

Author Response

Dear Reviewer,

Many thanks for your helpful comments. Please see the attachment (in pdf format) with our point to point reply to the comments. However, we do apologize for our long reply at comment 3 but, given the point raised by the reviewer, we found very useful to extend the response over this particular matter.

Round 2

Reviewer 2 Report

Accept in present form.